# Evaluating Sequence Alignment Tools for Antimicrobial Resistance Gene Detection in Assembly Graphs

**DOI:** 10.3390/microorganisms12112168

**Published:** 2024-10-28

**Authors:** Yusreen Shah, Somayeh Kafaie

**Affiliations:** Department of Mathematics and Computing Science, Saint Mary’s University, Halifax, NS B3H 3C3, Canada; yusreen.shah@smu.ca

**Keywords:** antimicrobial resistance (AMR), sequence alignment, assembly graphs, Bandage, SPAligner, GraphAligner

## Abstract

Antimicrobial resistance (AMR) is an escalating global health threat, often driven by the horizontal gene transfer (HGT) of resistance genes. Detecting AMR genes and understanding their genomic context within bacterial populations is crucial for mitigating the spread of resistance. In this study, we evaluate the performance of three sequence alignment tools—Bandage, SPAligner, and GraphAligner—in identifying AMR gene sequences from assembly and de Bruijn graphs, which are commonly used in microbial genome assembly. Efficiently identifying these genes allows for the detection of neighboring genetic elements and possible HGT events, contributing to a deeper understanding of AMR dissemination. We compare the performance of the tools both qualitatively and quantitatively, analyzing the precision, computational efficiency, and accuracy in detecting AMR-related sequences. Our analysis reveals that Bandage offers the most precise and efficient identification of AMR gene sequences, followed by GraphAligner and SPAligner. The comparison includes evaluating the similarity of paths returned by each tool and measuring output accuracy using a modified edit distance metric. These results highlight Bandage’s potential for contributing to the accurate identification and study of AMR genes in bacterial populations, offering important insights into resistance mechanisms and potential targets for mitigating AMR spread.

## 1. Introduction

Recent advancements in genome sequencing technology have significantly expanded our ability to study microbial communities at an unprecedented scale. With high-throughput sequencing, it is now possible to sequence entire microbiomes, leading to the availability of reference genomes and improved insights into microbial interactions. However, these technological breakthroughs come with their own set of challenges. The vast amount of data generated through short-read sequencing often provide fragmented DNA sequences, requiring sophisticated assembly methods to reconstruct longer, more meaningful genomic sequences [1]. These methods result in large assembly graphs, from which extracting precise genomic information is critical, particularly in understanding complex microbial behaviors such as the spread of antimicrobial resistance (AMR).

One of the most pressing issues in global health today is the rise of AMR [2] with an estimated 1.27 million deaths attributed to AMR in 2019 alone [3]. Horizontal gene transfer (HGT) of mobile genetic elements (MGEs), such as plasmids and genomic islands, can drive the spread of AMR between organisms and habitats [4]. An AMR gene’s associated neighbouring genes can play key roles in its function [5] and likelihood of undergoing HGT [6,7]. Identifying and tracking the presence of these AMR genes within microbial communities is a crucial step toward combating the crisis. This research focuses on evaluating state-of-the-art sequence alignment tools, particularly their efficiency and accuracy in aligning sequences within complex assembly graphs. These tools are indispensable for identifying AMR genes, their neighboring genes, and the genomic context in which horizontal gene transfer occurs, offering vital insights into how resistance genes spread.

This study aims to compare popular sequence alignment tools such as Bandage [8], SPAligner [9], and GraphAligner [10], all of which are commonly used in the analysis of de Bruijn graphs created during genome assembly. By evaluating these tools, this study seeks to provide a comprehensive analysis of their strengths and limitations with the goal of identifying the most efficient and accurate tool for the task.

The impact of this work extends beyond tool comparison. By improving the precision of AMR gene identification and mapping through better software selection, we contribute directly to the efforts tackling the AMR crisis. The insights from this study can enhance the detection of horizontal gene transfer events, thus informing public health strategies aimed at mitigating the spread of resistance.

## 2. Background

In genomic and metagenomic analyses, assemblers typically utilize de Bruijn graphs as a core structure to represent the relationships between DNA sequence reads. De Bruijn graphs are constructed from reads, which are sequences of base pairs corresponding to all or part of a single DNA (or multiple DNAs in case of metagenomic assembly). In this paradigm, the reads are broken into substrings of fixed length, known as *k*-mers. The nodes in the graph represent the k−1 prefixes and suffixes of these *k*-mers, while the edges correspond to the *k*-mers themselves. Instead of aligning reads directly, overlaps between sequences are inferred based on shared *k*-mers [11]. Many assemblers generate a simplified and compact version of the de Bruijn graph, known as the assembly graph, as the final output.

While there are many tools to identify AMR genes in large sequences such as contigs (i.e., a contiguous sequence of DNA that is assembled from an assembly graph), there are relatively few tools that can perform this task within an assembly graph. In an assembly graph, the sequence of an AMR gene might be spread across multiple connected nodes, representing a path in the graph. Traversing and identifying paths in such graphs is a more complex task than searching through linear sequences of contigs. In the following, we discuss three sequence alignment tools that can be used to identify AMR genes in an assembly graph.

Bandage [8] is primarily used for visualizing and interacting with assembly graphs. It also includes the ability to perform BLAST [12] searches directly within the graphical interface, which adds functionality to traditional BLAST by enabling searches through assembly graph nodes. Bandage builds a BLAST database from all the nodes in the graph, and for each query, it attempts to find a path through the graph that maximizes the coverage of the query. In addition to visualization, Bandage’s integration of BLAST functionality makes it a versatile tool for graph-based sequence analysis.

Saint Petersburg Aligner (SPAligner) [9] is a tool designed for aligning nucleotide and amino acid sequences against assembly graphs. SPAligner identifies regions of high nucleotide identity between the query sequence and the graph and extends these regions into semi-global alignments. In semi-global alignment, gaps are allowed at the beginning and/or the end of the sequence, making it suitable for aligning sequences that may only overlap partially. If two sequences are related over the entire length of their overlapping regions, semi-global alignment is used to determine the optimal alignment. SPAligner has proven to be efficient in handling this type of alignment task within assembly graphs.

GraphAligner [10] is another alignment tool that specializes in aligning reads to sequence graphs. GraphAligner can work with a variety of graph types and is particularly well-suited for handling the complex structures found in long-read sequencing data. By aligning reads directly to the graphs, GraphAligner facilitates the identification of complex structural variants and improvements in assembly accuracy.

In this study, we compare these tools—Bandage, SPAligner, and GraphAligner—focusing on their ability to accurately detect sequences in assembly graphs, particularly in the context of antimicrobial resistance (AMR) gene detection. All three tools utilize de Bruijn graphs in their analysis. Specifically, Bandage accepts de novo assembly graphs in LastGraph (Velvet), FASTG (SPAdes), Trinity.fasta, ASQG, and GFA formats. GraphAligner works with genome graphs in GFA and VG formats, while SPAligner accepts assembly graphs in GFA format. We aim to evaluate their performance in terms of accuracy and computational efficiency, providing insights into their respective strengths and limitations. Table 1 summarizes the key points about these tools. As shown in the table, Bandage provides a way to visualize the graph as well as sequence alignment. While SPAligner and GraphAligner have similar usage in terms of sequence alignment, they use different algorithms under the hood for this purpose.

## 3. Materials and Methods

In this study, we evaluate the performance of three sequence alignment tools—Bandage (v0.9.0), SPAligner (v3.15.5), and GraphAligner (v1.0.17)—for aligning sequences in assembly graphs. We compare these tools in terms of time and memory usage. Additionally, we assess the accuracy of the output sequences identified by each tool across various datasets, as summarized in the flowchart in Figure 1.

Each tool receives the assembly graph (in GFA [13] format) generated by tools such as metaSPAdes [14], BCALM [15], or megahit [16]. In these assembly graphs, fragments of DNA sequences are presented as nodes (i.e., segments), while overlapping segments are connected via edges. Each tool also receives a list of AMR genes as input. The tools are expected to return the path(s) representing each AMR gene (if identified) within the graph. For example, in Figure 2, the query sequence is identified as a path starting from the 90th nucleotide at node n2 and will end at the 80th nucleotide at node n3. By analyzing the output generated by Bandage, SPAligner, and GraphAligner, we have identified several common features for comparison. These include:The length of the query sequence;The starting and ending positions of the aligned sequence in the graph;The specific nodes involved in the path through the graph;The identified output sequence.

These shared characteristics serve as the basis for evaluating the performance and effectiveness of the tools in identifying sequences within genome assembly graphs.

### 3.1. Dataset

As detailed in Table 2, we conducted experiments on assembly graphs of varying sizes generated by metaSPAdes v3.14.1 [14] from the following datasets which range from small to large:1_1_1: A simulated metagenomic dataset generated using one strain each from *Escherichia coli*, *Staphylococcus aureus*, and *Klebsiella pneumoniae*, retrieved from RefSeq. The reads were simulated using ART V2.5.8 [17] on the HiSeq 2500 platform with a read length of 150 bp, an insert size of 500 bp, and fold coverage of 20.CAMI_M_2 and CAMI_H_1: These datasets are part of the Critical Assessment of Metagenome Interpretation (CAMI) [18] study and represent medium and large-sized assembly graphs, offering real-world challenges for metagenome analysis.ERR1713331: A metagenomic dataset derived from urban sewage from Albania, published and sequenced using the Illumina HiSeq platform [19].

### 3.2. Experiment

For each dataset, AMR sequences identified in the dataset were used as queries. The three tools—Bandage, SPAligner, and GraphAligner—were executed to find these AMR sequences within the respective assembly graphs. All experiments were run on an iMac with an Apple M1 chip and 8 GB of memory.

**Table 2 microorganisms-12-02168-t002:** Datasets simulated from RefSeq (1_1_1), the CAMI Challenge [18], and a real metagenomic sample selected from the Global Urban Sewage AMR Monitoring Project [19].

Name	Description	Number of AMR Genes	Graph Size
**1_1_1**	Simulated from *E. coli* SMS-3-5 (NC_010498, NC_010488, NC_010485, NC_010486, NC_010487), *K. pneumoniae* MGH 78578 (NC_009648, NC_009649, NC_009650, NC_009651, NC_009652, NC_009653), *S. aureus* Mu50 (NC_002758, NC_002774) (accessed on 8 October 2024)	378	2529 nodes and 2406 edges
**CAMI_M_2**	CAMI Challenge with 132 genomes	54	396,319 nodes and 101,235 edges
**CAMI_H_1**	CAMI Challenge with 596 genomes	698	939,234 nodes and 127,706 edges
**ERR1713331**	Albania (ERR1713331) (accessed on 8 October 2024)	355	3,852,226 nodes and 1,256,367 edges

For the experiments, default parameter values were used for all tools, except for *minmeanid* in Bandage. The default *identity* threshold for GraphAligner was set to 0.66, so to maintain consistency and ensure a fair comparison, we adjusted the *minmeanid* value for Bandage to 0.66. We compared the results from each tool in three key areas:Time: The total time taken for each tool to align the query sequences.Memory Usage: The memory footprint of each tool during the alignment process.Accuracy: The accuracy of the alignment was assessed by comparing the edit distance between the query sequence and the sequences returned by each tool.

The accuracy metric provides a quantitative measure of how well each tool aligns the input query sequences within the assembly graph, helping us determine their overall performance in handling genome assembly tasks involving AMR gene detection.

## 4. Results

In this section, we present the analysis of the results from Bandage, SPAligner, and GraphAligner. The analysis is structured into three main subsections:Path Comparison: We compare the paths returned by the three tools, focusing on the start and end positions as well as the nodes involved in each path for all datasets.Time and Memory Consumption: We compare the time and memory usage of each tool across the different datasets.Accuracy Evaluation: We assess the accuracy of the sequences returned by each tool by comparing them to the query sequences.

### 4.1. Path Comparison

For each dataset, we combined the query sequences into a single file. This file, along with the corresponding assembly graph, was used as input for each tool. The outputs were compared pairwise by examining the start position, end position, and the list of nodes involved in the paths.

The comparison of paths returned by each pair of tools was categorized as follows:Full: The start position, end position, and node list are identical in the paths returned by both tools.Partial: The paths returned by the tools show some meaningful similarities but differ in either the start or end position (or both), while still covering similar regions of the graph.Different: The paths differ entirely in terms of start position, end position, and the nodes involved.

The results of the comparison for each dataset, as shown in Table 3, Table 4, Table 5 and Table 6 demonstrate that Bandage and GraphAligner return the most similar outputs.

### 4.2. Time Comparison

Figure 3 provides a visual representation of the time taken by each tool for sequence alignment. As shown in this figure, Bandage consistently took the shortest time to align all sequences across all datasets. With the exception of the 1_1_1 dataset, SPAligner generally required the longest time to complete sequence alignment.

### 4.3. Memory Consumption

Memory usage across the tools is displayed in Figure 4. Bandage consistently used the smallest amount of memory in all cases. Except for the 1_1_1 dataset, GraphAligner consumed the most memory among the tools. The memory consumption for each tool during the alignment process is shown in the plots below.

### 4.4. Measuring Match Rate

To evaluate the quality of the sequences produced by each tool, we measured the similarity between the target query sequence and the output sequences generated by the three tools. This similarity was quantified using the edit distance, which counts the minimum number of changes (insertions, deletions, or substitutions) required to transform one sequence into another [20].

To adapt the edit distance metric to our specific needs, we normalized it by dividing it by the length of the sequences (specifically, the maximum length of the two sequences). Additionally, since we are focusing on sequence similarity rather than dissimilarity, we subtract the normalized value from 1. This adjustment ensures that a higher value corresponds to greater similarity between sequences. This new metric, referred to as match_rate, was calculated using the following formula: (1)$$match\_rate=1-\frac{edit\_Distance}{max(length(targetSequence),length(output))}$$

The closer the match_rate is to 1, the more similar the output sequence is to the target query sequence. For queries where Bandage returned multiple paths and sequences, we selected the path with the highest confidence to ensure consistency in the comparison.

As shown in Figure 5, the percentage of sequences with different match_rate values for each tool and dataset demonstrates that Bandage achieved the highest match_rate. This indicates that Bandage outperforms the other two tools, frequently extracting sequences from the graph that were the closest to the query AMR sequences (in many cases, identical). In the 1_1_1 dataset, GraphAligner produced the second most accurate sequences. However, for the CAMI_M_2 and real datasets, SPAligner outperformed GraphAligner in terms of accuracy. For the CAMI_H_1 dataset, both Bandage and GraphAligner returned sequences with match_rates mostly around 0.5, indicating a moderate level of accuracy. In contrast, SPAligner produced a significant number of sequences that did not match the target sequences at all.

Figure 6 presents the average match_rate of each tools across all datasets, further emphasizing Bandage’s superior performance.

## 5. Conclusions and Future Work

In this study, we evaluated the performance of three widely used sequence alignment tools—Bandage, GraphAligner, and SPAligner—with a focus on their ability to detect antimicrobial resistance (AMR) genes within assembly graphs. To evaluate the performance of these tools, we reconstructed sequences from the identified paths and compared them with the original AMR gene sequences. We assessed their performance in terms of accuracy, run-time, and memory consumption across datasets of varying sizes, ranging from small (1_1_1) to medium (CAMI_M_2) to large (CAMI_H_1 and real-world datasets). To measure the accuracy of their results, we introduced the match_rate metric, which quantifies the similarity between the query sequence and the output sequence.

Comparison of the accuracy of each tool, using the match_rate metric, confirms that Bandage offers the most accurate identification of AMR genes, with sequences most closely matching the AMR gene queries. Additionally, Bandage proved to be the most efficient in terms of both time and memory usage, making it the best overall tool for AMR gene detection in assembly graphs. GraphAligner generally followed Bandage in performance, especially in terms of run-time, while SPAligner lagged behind, particularly in larger datasets.

In terms of accuracy, GraphAligner outperformed SPAligner in two of the datasets, although it required additional coding effort to extract the output sequences. On the other hand, SPAligner automatically constructs sequences but missed some AMR gene queries, reducing its overall reliability for accurate sequence alignment.

For future research, exploring newer tools that are emerging in the field of sequence alignment may provide valuable alternatives to Bandage, potentially offering more comparable results than those of GraphAligner and SPAligner. These tools could further enhance the detection of AMR genes in complex assembly graphs, contributing to more effective genomic analysis in combating antimicrobial resistance.

Furthermore, as sequencing technologies evolve, particularly with the rise of long-read sequencing platforms, future work could focus on optimizing existing alignment tools or developing new ones that can efficiently handle these longer reads within assembly graphs. The ability to accurately align and analyze longer reads may lead to more detailed reconstructions of microbial genomes and better tracking of AMR gene dissemination.

Lastly, expanding the datasets used for benchmarking tools to include more diverse and complex real-world samples could provide more comprehensive insights into the tools’ robustness. These future experiments could focus on a wider range of metagenomic environments, including those with higher variability in microbial content, to better reflect the challenges encountered in real-world AMR monitoring and research.

## Figures and Tables

**Figure 1 microorganisms-12-02168-f001:**
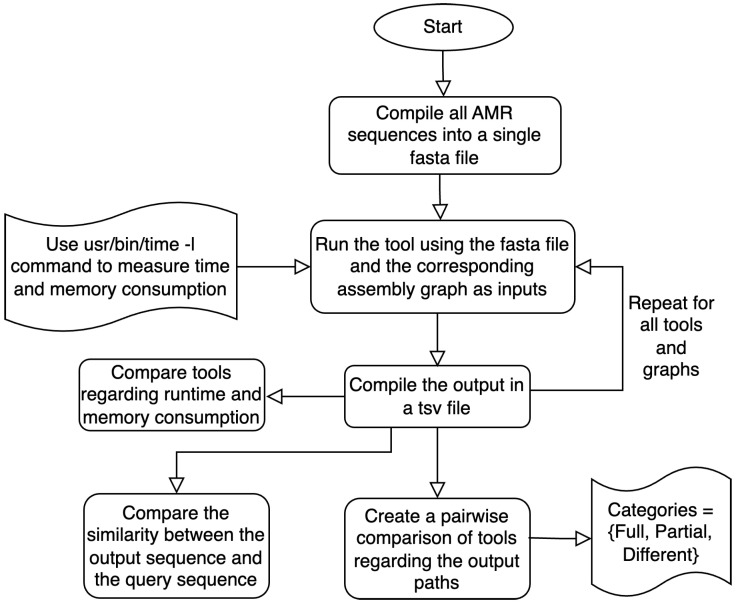
This flowchart shows the steps used to compare the results from the output file for each graph.

**Figure 2 microorganisms-12-02168-f002:**
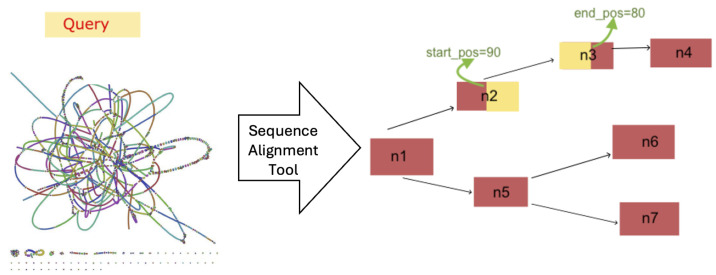
Sequence alignment in the assembly graph for a query sequence, where the yellow part in the graph represents the output path found by a given tool for the query.

**Figure 3 microorganisms-12-02168-f003:**
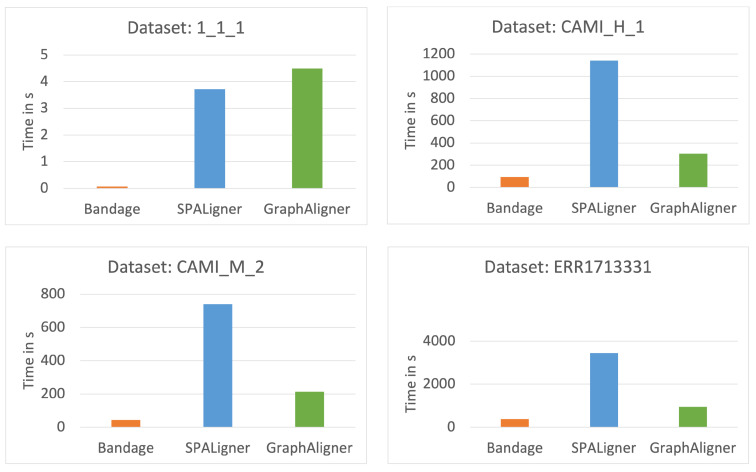
Execution time of the tools for each dataset.

**Figure 4 microorganisms-12-02168-f004:**
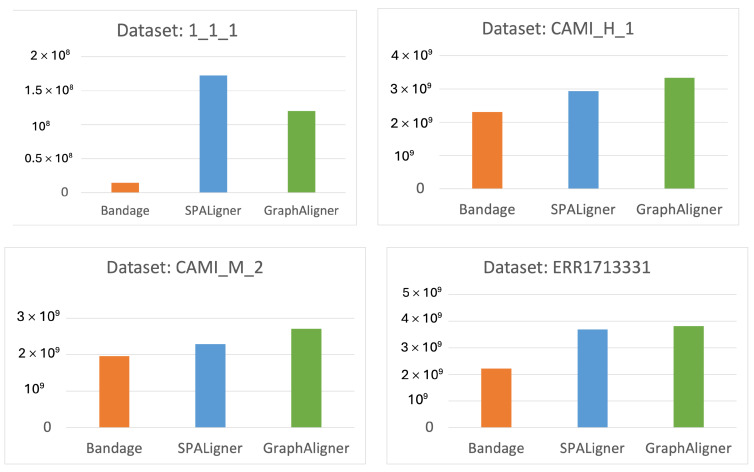
Memory consumption of the tools for each dataset.

**Figure 5 microorganisms-12-02168-f005:**
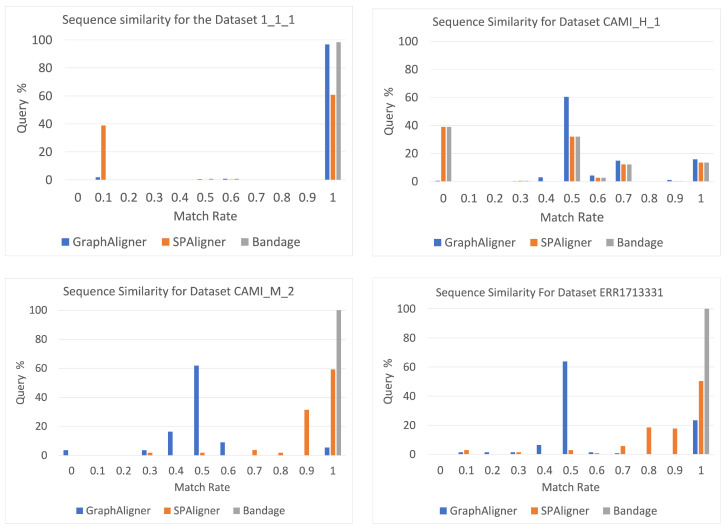
The percentage of sequences with different values of match_rate for each software and dataset.

**Figure 6 microorganisms-12-02168-f006:**
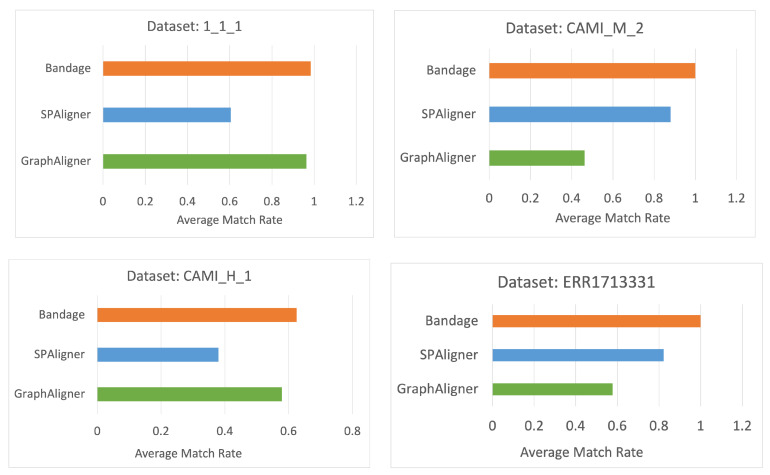
The average match_rate for each software and dataset.

**Table 1 microorganisms-12-02168-t001:** This table describes the differences between the three tools in terms of usage and algorithms.

Tool	Main Usage	Alignment Algorithm
Bandage	Primarily to visualize and interact with assembly graphs	Uses BLAST to find paths within the assembly graph based on the query sequence.
SPAligner	Align diverged molecular sequences against assembly graphs	Uses Burrows–Wheeler aligner (BWA) to detect longer anchor alignments. It also uses the Edlib library to calculate the optimal alignment.
GraphAligner	Align long reads to sequence graphs	Uses BWA to detect longer anchor. It also uses the bit vector alignment extension algorithm.

**Table 3 microorganisms-12-02168-t003:** The percentage of each category for 1_1_1 in pair comparison of the tools.

Tool Pair	Full	Partial	Different
Bandage VS SPAligner	57.9	7.2	34.9
Bandage VS GraphAligner	93.9	2.1	4.0
SPAligner VS GraphAligner	57.7	4.8	37.6

**Table 4 microorganisms-12-02168-t004:** The percentage of each category for CAMI_M_2 in pair comparison of the tools.

Tool Pair	Full	Partial	Different
Bandage VS SPAligner	83.3	0	16.7
Bandage VS GraphAligner	63	20.3	16.7
SPAligner VS GraphAligner	0	85.2	14.8

**Table 5 microorganisms-12-02168-t005:** The percentage of each category for CAMI_H_1 in pair comparison of the tools.

Tool Pair	Full	Partial	Different
Bandage VS SPAligner	0	94.5	5.5
Bandage VS GraphAligner	71.0	24.0	5.0
SPAligner VS GraphAligner	0	95.3	4.7

**Table 6 microorganisms-12-02168-t006:** The percentage of each category for real sample in pair comparison of the tools.

Tool Pair	Full	Partial	Different
Bandage VS SPAligner	0	63.8	36.2
Bandage VS GraphAligner	22.1	23.5	54.4
SPAligner VS GraphAligner	0	63.1	36.9

## Data Availability

The code and data supporting this work are available at https://github.com/kafaie-lab/Comparison_sequenceAligners accessed on 14 October 2024.

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
