# Peer review of "Evaluating Sequence Alignment Tools for Antimicrobial Resistance Gene Detection in Assembly Graphs"

_microorganisms, 2024, doi:10.3390/microorganisms12112168_

Round 1

Reviewer 1 Report

Comments and Suggestions for Authors

The monitoring and prevention of antimicrobial resistance (AMR) is a global challenge. This study aims to compare popular sequence alignment tools, namely Bandage, SPAligner, and GraphAligner, which are commonly utilized in the analysis of de Bruijn graphs generated during genome assembly. By assessing these tools, the research endeavors to provide a thorough analysis of their strengths and limitations, ultimately aiming to identify the most efficient and accurate tool for the task.

(1) The information presented in Table 1 lacks refinement and has already been covered in the main text. It is recommended to further distill the core information and present it in a condensed form; otherwise, there is no need for the table.

(2) Is Figure 1 original? If not, please provide the original reference source.

(3) The meaning of the terms in Formula 1 needs further verification as it is currently unclear. Could you please clarify where this information was obtained?

(4) The future outlook section is too brief. It is recommended to expand and elaborate on this section.

Author Response

The monitoring and prevention of antimicrobial resistance (AMR) is a global challenge. This study aims to compare popular sequence alignment tools, namely Bandage, SPAligner, and GraphAligner, which are commonly utilized in the analysis of de Bruijn graphs generated during genome assembly. By assessing these tools, the research endeavors to provide a thorough analysis of their strengths and limitations, ultimately aiming to identify the most efficient and accurate tool for the task.]

Response: We sincerely thank the reviewer for acknowledging the value and contributions of our manuscript. We also appreciate the constructive feedback, which has greatly helped in improving the overall quality of the manuscript. We have carefully considered all specific comments and provide detailed responses to each below.

Comment 1: The information presented in Table 1 lacks refinement and has already been covered in the main text. It is recommended to further distill the core information and present it in a condensed form; otherwise, there is no need for the table.

Response 1: Thank you for your insightful comment. We agree that the information in the second column (“Main Usage”) has already been covered in the text, which is why we aimed to provide a brief summary in the table. However, we would like to emphasize that the third column (“Alignment Algorithm”) contains additional details that are not explicitly discussed in the main text. We hope this clarification justifies the inclusion of the table, as it provides new, valuable information.

Comment 2:  Is Figure 1 original? If not, please provide the original reference source.

Response 2: Thank you for your query. We confirm that Figure 1 is original and was specifically created to illustrate the tasks completed during this research. To enhance clarity, we have revised the figure in the updated manuscript.

Comment 3:  The meaning of the terms in Formula 1 needs further verification as it is currently unclear. Could you please clarify where this information was obtained?

Response 3: We appreciate the opportunity to clarify this point. The edit distance metric used in Formula 1 is a well-established measure, initially introduced in reference [20]. To adapt it to our specific needs, we normalized the edit distance by dividing it by the length of the sequences (specifically, the maximum length of the two sequences). Additionally, since we are focusing on sequence similarity rather than dissimilarity, we subtract the normalized value from 1. This adjustment ensures that a higher match rate corresponds to greater similarity between sequences. We have added this explanation to the manuscript to ensure clearer understanding.

Comment 4The future outlook section is too brief. It is recommended to expand and elaborate on this section.

Response 4: Thank you for this recommendation. In response, we have expanded the future outlook section in the revised manuscript, providing more detailed discussion on potential future directions and areas for further research.

Reviewer 2 Report

Comments and Suggestions for Authors

This is a clearly written manuscript that perhaps falls outside the scope of this journal. The authors compare three tools to identify antibiotic resistance genes. While I was initially enthusiastic, the manuscript went on to compare the three tools, but the main goal, which tool was better, was never clear or easily provided by the authors.

For example, if I subject a K. pneumoniae whole genome sequence that has 378 known antibiotic resistance genes, 378,2529 nodes, and 2406 edges, which of the three tools was closer to these values?

How are these three tools different from programs already available? Are targeted assemblers better, or what will you do with tools such as CARD, Staramr, ABRicate, and ResFinder? Although computing time is important, I would love to see clearly which of the three tools analyzed provided the best and most accurate identification of antibiotic-resistance genes provided in Table 2.

Are these three programs/tools using overlap–layout–consensus (OLC) and de-Bruijn-graph as part of their analysis?

Author Response

We sincerely thank the reviewer for their positive feedback and constructive comments, which have greatly contributed to improving the quality of our manuscript. We have carefully considered all the points raised and made the necessary revisions to the manuscript. Below are our detailed responses to each comment.

Comment 1: This is a clearly written manuscript that perhaps falls outside the scope of this journal. The authors compare three tools to identify antibiotic resistance genes. While I was initially enthusiastic, the manuscript went on to compare the three tools, but the main goal, which tool was better, was never clear or easily provided by the authors. For example, if I subject a K. pneumoniae whole genome sequence that has 378 known antibiotic resistance genes, 378,2529 nodes, and 2406 edges, which of the three tools was closer to these values?

Response 1: Thank you for your insightful comment. The three tools discussed in the manuscript (Bandage, GraphAligner, and SPAligner) are capable of identifying the paths in the assembly graph that represent specific AMR genes. To evaluate the performance of these tools, we reconstructed sequences from the identified paths and compared them with the original AMR gene sequences. For this purpose, we introduced the match_rate metric, which quantifies the similarity between the query sequence and the output sequence. As shown in Figure 5, the percentage of sequences with different match_rate values for each tool and dataset demonstrates that Bandage outperforms the other two tools, frequently finding an exact match in the assembly graph. Similarly, Figure 6 presents the average match_rate for each tool and dataset, further emphasizing Bandage’s superior performance. We have revised the manuscript to make this conclusion clearer.

Comment 2: How are these three tools different from programs already available? Are targeted assemblers better, or what will you do with tools such as CARD, Staramr, ABRicate, and ResFinder?

Response 2: Thank you for raising this important point. While tools like CARD, Staramr, ABRicate, and ResFinder can indeed identify AMR genes in large sequences such as contigs, there are relatively few tools (e.g., Bandage, GraphAligner, and SPAligner) that can perform this task within an assembly graph. In an assembly graph, the sequence of an AMR gene might be spread across multiple connected nodes, representing a path in the graph. Traversing and identifying paths in such graphs is a more complex task than searching through linear sequences of contigs. We have revised the manuscript to clarify this distinction and to highlight the unique capabilities of the tools evaluated in this study.

Comment 3: Although computing time is important, I would love to see clearly which of the three tools analyzed provided the best and most accurate identification of antibiotic-resistance genes provided in Table 2.

Response 3: Thank you for your comment. As mentioned in our response to Comment 1, comparing the accuracy of the three tools is one of the primary objectives of this study. Figures 5 and 6 provide a clear comparison of the accuracy of each tool using the match_rate metric, confirming that Bandage offers the most accurate identification of AMR genes. We have ensured that this is more explicitly stated in the revised manuscript.

Comment 4: Are these three programs/tools using overlap–layout–consensus (OLC) and de-Bruijn-graph as part of their analysis?

Response 4: Thank you for your question. All three tools utilize de Bruijn graphs in their analysis. Specifically, Bandage accepts de novo assembly graphs in LastGraph (Velvet), FASTG (SPAdes), Trinity.fasta, ASQG, and GFA formats. GraphAligner works with genome graphs in GFA and VG formats, while SPAligner accepts assembly graphs in GFA format. We have clarified these points in the revised manuscript to address this query.